# Fluoride Concentration in Urine after Supplementation with Quelites in a Population of Adolescents

**DOI:** 10.3390/foods11193071

**Published:** 2022-10-03

**Authors:** Yair Olovaldo Santiago-Saenz, Rebeca Monroy-Torres, Diana Olivia Rocha-Amador, César Uriel López-Palestina

**Affiliations:** 1Área Académica de Nutrición, Instituto de Ciencias de la Salud, Universidad Autónoma del Estado de Hidalgo, San Agustín Tlaxiaca 42160, Hidalgo, Mexico; 2Laboratorio de Nutrición Ambiental y Seguridad Alimentaria, Departamento de Medicina y Nutrición, División de Ciencias de la Salud, Universidad de Guanajuato, Campus León, Predio de San Carlos s/n, León 37670, Guanajuato, Mexico; 3Departamento de Farmacia, División de Ciencias Naturales y Exactas, Universidad de Guanajuato, Campus Guanajuato, Guanajuato 36050, Guanajuato, Mexico; 4Área Académica de Ingeniería Agroindustrial e Ingeniería en Alimentos, Instituto de Ciencias Agropecuarias, Universidad Autónoma del Estado de Hidalgo, Tulancingo de Bravo 43600, Hidalgo, Mexico

**Keywords:** fluoride, supplementation, quelites, water, metals

## Abstract

Wild plants have detoxifying and protective properties. They reduce or neutralize the toxic effects caused by chemical agents and pollutants and have beneficial effects on the nutritional and health status. This research was conducted to complement a previous clinical intervention in which participants were exposed to high concentrations of arsenic and fluoride in drinking water to discover similarities or differences in the pattern of fluoride (F^−^) excretion after supplementation with partial areas of purslane and quelite cenizo (SP-Q). The study was 4 weeks long, and it was carried out on a sample of 27 adolescents in an age range of 11–12 years. Anthropometric and dietary evaluations were performed, and the concentrations of fluoride (F^−^) in urine and drinking water were quantified using the potentiometric method with an ion-selective electrode. The treated group increased urinary F^−^ excretion after the first week (Baseline: 9.23 mg/g creatinine; Week 2: 0.73 mg/g creatinine), together with significant mobilization of F^−^ and a recovery process at the end of the intervention (Week 4: 0.52 mg/g creatinine). The supplement may act by increasing the excretion of F^−^ and the nutritional dietary conditions that contribute to mitigation and recovery in participants exposed to the contaminant while also managing access to drinking water.

## 1. Introduction

Two of the most important contaminants recognized internationally are arsenic (As) and the fluoride ion (F^−^), which are frequently found bound in water [1,2]. They are inorganic contaminants of water resources destined for productive activities and for daily human consumption, such as drinking water or water for food preparation [3,4]. The presence of these chemical compounds has been reported in the Asian continent and in Latin American countries, such as Argentina and Mexico, in concentrations that fluctuate in ranges from 0.5 to 5000 µg/L [5] and from 0.5 to 2800 mg/L [6,7] for As and F^−^, respectively. Some states in Mexico have been identified as areas that contain amounts greater than the established limits of As and F^−^, mainly due to natural contamination. Among these are the cities of San Luis Potosí, Zacatecas, Sonora, Guanajuato, Chihuahua, Coahuila, and Hidalgo [5,8,9,10,11,12,13,14]. There have been investigations on the presence of As and F^−^ in the water conducted in municipalities of the state of Guanajuato, which, due to its geographical location and geological characteristics, contains toxic elements [5,13,15]; Regarding F^−^, its presence in groundwater reserves has been associated with rocks that present quartz, feldspars, fluorite, and apatite [1,13]. For this reason, pollution of natural and anthropogenic origin in geological environments is a risk factor for the availability of water of sufficient quality for human consumption, as the water supply is mostly provided by sources of underground origin [1].

As and F^−^ are elements known for their toxicity at certain concentration levels and which present a consequent risk to public health [4,5,9,10]. Regarding F^−^, it is one of the most reactive compounds that belong to a wide group of chemical compounds that are commonly found in water, soil, and air and which can be transported to food chains [4]. On the other hand, F^−^ is considered a trace element with essential biochemical functions since it is required for the formation of bone tissue and for the maintenance of bone integrity [4,14]. F^−^ is added in small amounts to commonly used health products, such as toothpaste and medications, including intentionally addition to water to prevent dental caries [4,16]. However, being classified as a potentially toxic element, exposure to high levels of this ion can harm health [4,14]. Additionally, prolonged exposure to F^−^ in drinking water can cause dental and skeletal fluorosis as well as adverse neurological and reproductive effects [4,7,8,9,17,18]. In Mexico, there has been an increase in the presence of fluorosis in children due to excessive intake of this element in drinking water, which has increased fluoride retention and caused dental fluorosis in these age groups [14]. Therefore, the contamination of aquifers by the presence of F^−^ constitutes a national public health problem and is a cause for concern. Thus, in addition to risk communication strategies, it is necessary to exercise the right to clean water quality and implement other nutritional intervention alternatives, such as the consumption of freeze-dried endemic plants of the region (quelites) or the intake of vitamin supplements previously prepared with species rich in nutrients [5,10]. Possible coadjuvants in the process of F^−^ excretion promote a reduction in the concentration of these toxic components and, therefore, their toxic effects on the body. Various studies have reported that a diet rich in phytochemicals can prevent some of the processes involved in the development of cancer and cardiovascular and chronic diseases and can neutralize or detoxify toxins, improving the antioxidant capacity and providing protective effects against lipid oxidation caused by pollutants, drugs, fertilizers, and other factors that compromise human health [19,20,21]. There is evidence that the intake of foods rich in antioxidants and nutrients, such as phosphorus, calcium, magnesium, and vitamins C, E, and D, through incorporation into the diet through the consumption of fruits, vegetables, or supplements, can decrease the toxic effects of various metals and metalloids, such as As and F^−^, and influence the toxicokinetics of these elements. The study by Monroy-Torres, et al. [10] showed that multivitamin supplementation plus nutritional care increased arsenic excretion [10,22,23,24,25].

Currently, in Mexico, there is access to a wide variety of endemic plants known as quelites, which are of wild origin and are adaptable to any type of climate in the Mexican Republic, making their acquisition relatively easy [26]. Various investigations have reported that the species of interest in this study (*Chenopodium berlandieri* L. and *Portulaca oleraceae* L.) contain a wide variety of phenolic compounds (chlorogenic acid, phloridzin, naringenin, and phloretin) and macro and micronutrients (Ca, Mg, K, Fe, Mn, and amino acids), among other phytochemical compounds (carotenoids and chlorophyll) [27,28,29]. Quelites, due to their nutritional richness, have been reported to have beneficial properties for human health, including antioxidant, anti-inflammatory, and anticancer effects. Additionally, they promote the reduction of damage caused by exposure to pesticides such as rotenone [28,30,31]. On the other hand, according to Santiago et al. [5], who studied the effect of the same supplement based on purslane and quelite cenizo in a population exposed to As in drinking water, the participants in the intervention group showed increased excretion of arsenic (2.02 vs. 56.49 µg/g creatinine for the treated and control groups, respectively) and a reduced urine MDA concentration (1.59 vs. 2.90 µM/g creatinine for the treated and control groups, respectively) after the second week of supplementation. This evidence supports the consumption of native Mexican plants to act against various oxidative stress processes.

In vitro and in vivo studies have been conducted to investigate supplementation with vitamins and inorganic nutrients with different presentations. Monroy-Torres et al. [10] observed an effect on the excretion of urinary arsenic with the supplementation of vitamins and minerals plus the nutritional intervention, and the methodology of this study was considered for the study of Santiago-Saenz et al. [5]. However, it is important to explore other sources of these key nutrients (quelites) with a clinical design to investigate the elimination or detoxification of F^−^ from the body. Therefore, the objective of this research was to evaluate the effectiveness of a vegetable supplement containing quelites (SP-Q) on the level of F^−^ excretion in participants aged 11–12 years exposed to water with F^−^ and belonging to the rural community of Guanajuato, Mexico.

## 2. Materials and Methods

### 2.1. Study Design

The intervention study was carried out in a rural area known as Valencianita, belonging to the city of Irapuato in the State of Guanajuato, Mexico. This community has previously been reported as a geographic area with high levels of arsenic (As) and fluoride ions (F^−^) in water [10]. This study corresponds to a secondary objective that was developed within the same evaluation period as a clinical research project [5] pertaining to arsenic, which was submitted, evaluated, and approved by an Institutional Ethics Committee “*The Institutional Bioethics Committee of the University of Guanajuato” under register number ID: CIBIUG-P34-2018*. The intervention presented was a randomized clinical study (Figure 1). Based on the inclusion criteria, the participants were called (Parents and children) and provided with verbal and written information about the objectives of the study. A list of interested participants was obtained. Subsequently, the inclusion of the interested parties was confirmed after they had signed the corresponding consent and assent forms voluntarily, in accordance with the respective selection criteria [5]. F^−^ concentrations in drinking water of greater than 1.5 mg/L were considered (hydrofluorosis) [32]. Subsequently, the codes were assigned by randomization using the SAS^®^ program (Version 9.4, SAS Institute, NC, USA). Finally, the participants were assigned to two groups: group 1 (G1-Treated) and group 2 (G2-Untreated). The distribution of the study participants was the same as that described by Santiago-Saenz et al. [5]. The baseline characteristics (age, weight, height, fat mass, waist circumference, and abdominal circumference) of the study groups can be observed in more detail in the paper published by Santiago-Saenz et al. [5]. Finally, to obtain better results, parents and study participants were instructed to reduce their consumption of piped water.

The sample size was calculated using the following formula: Sample size = [2 SD^2^ (Z_α/2_ + Z_β_)^2^]/d^2^(1)
where SD: Standard deviation (9); Z_α/2_: Confidence interval (95%); Z_β_: Statistical power (80%); d: Effect size (8).

The sample size was adjusted using the following formula:Adjusted sample size = *n* (1/1 − R)(2)
where *n*: Sample size before adjustment (20 per group); R: dropout proportion (17%).

### 2.2. Ethical Principles for Clinical Research (Volunteers)

All fundamental rights and general principles set forth in the Nuremberg Code and established in the Declaration of Helsinki were respected. The fundamental ethical principles of beneficence, non-maleficence, justice, and autonomy (60–62) were respected, following the guidelines of NOM-012-SSA3-2012, which establishes the criteria for the execution of research projects for human health. The regulation of the general health law DOF 04-02-2014 on health research was taken into account. Additionally, this study was conducted in accordance with the principles established in article 100 of the general health law DOF 06-01-2016 and in article 11 of resolution number 8430 of 4 October 1993 of the Ministry of Health. This investigation was considered to be of minimal risk and in compliance with the aspects mentioned in article 6 of this resolution. The researchers made sure that every participant was safe. The informed assent (minor) and consent (parent) processes were carried out. We presented all the details of the study and informed participants that they were able to leave the trial at any time and for any reason without being judged or put in a difficult position. Researchers kept personal information in a private location.

### 2.3. Supplementation

The supplement included in the clinical intervention was based on purslane and quelite cenizo (SP-Q). It was previously characterized, and its nutritional and antioxidant properties were evaluated independently and in a mixture using the same procedure as described in previous studies [27,29]. The quantity of dietary supplement consumed by participants was 4 g per day for 4 weeks, in accordance with Santiago Saenz et al. [5].

### 2.4. Anthropometric Assessment

The indicators were evaluated weekly according to the technique of Lohman et al. [33]. Waist and abdomen circumferences (cm) were measured using metallic tape (Lufkin-7 mm). Height (cm) (stadimeter, Seca^®^, Seca Mexico, Ciudad de Mexico, Mexico), weight (Kg), body fat (%), and fat-free mass (%) were measured using the impedance technique (bioimpedance scale, InBody^®^ R20, InBody Mexico, Ciudad de Mexico, Mexico). BMI was considered, and a diagnosis was established for the heights and weights of the participants using the Z-score-standard deviation (SD) for individuals aged 5–19 years. WHO reference tables (2007) and WHO Anthro Plus^®^ (Version 1.0.4, WHO, Geneva, Switzerland) software were used for data interpretation.

### 2.5. Dietary Evaluation

Data collection was performed weekly using the standardized food frequency survey in conjunction with a 24-h recall [34]. The procedure and the quantified nutrients used were described by Santiago-Saenz et al. [5]. For the analysis, the nutritional software NutriKcal^®^ (Version 1.1, Ogali-Consinfo, Ciudad de Mexico, Mexico) was used.

### 2.6. Chemicals and Reagents

Nitric acid, sodium citrate dihydrate, glacial acetic acid, sodium hydroxide, sodium chloride, and ethylenediaminetetraacetic acid (EDTA) were purchased from J.T Baker (Avantor Performance Materials, Ecatepec, Estado de Mexico, Mexico) and from Fermont (Rye S.A de C.V, Tlalnepantla, State of Mexico, Mexico). Standards for the F^−^ curve were purchased from Agilent (Agilent, Santa Clara, CA, USA). Quality controls for F^−^ (NIST-Fluoride in lyophilized urine, SRM 2671a) were purchased from Iris Tech (IRIS Technologies International GmbH, Olathe, KS, USA) and NIST (National Institute of Standards and Technology, Gaithersburg, MD, USA), respectively. Deionized water was used in all experiments and solutions.

### 2.7. Preparation of Urine Samples for Analysis of F^−^

The process of collecting, storing, and processing the samples for the analysis of F^−^ was as described by Santiago-Saenz et al. [5]. Polyethylene containers were used to collect the samples (first-morning urine); these were washed with nitric acid (10%). The samples were stored under refrigeration at 4 °C for later analysis. The creatinine content of the samples was determined by the Jaffe reaction method [35]. Urinary biomarker results were adjusted for the g of creatinine.

### 2.8. Determination of Fluoride in Water and Urine

To quantify fluoride ions (F^−^) in water, a potentiometer with a selective electrode (Crison instruments S.A, 9655, Barcelona, Spain) was used in accordance with method 3808 of the National Institute for Occupational Safety and Health of the United States (NIOSH) [36]. Samples were mixed with Total Ionic Strength Adjustment Buffer (TISAB) in a 1:1 ratio. Urine samples were added to 0.3 g of EDTA for every 100 mL of the sample following the previous methodology. Finally, the concentration of F^−^ was determined by interpolation of the potential in the calibration curve. As a quality control, the reference material SRM 2671a (NIST) was used with accuracy levels of 99% ± 2% and 98% ± 3% for urine and water, respectively. Measurements were made weekly. The results are expressed as mg F^−^/L and mg F^−^/g creatinine.

### 2.9. Statistical Analysis

The statistical analysis considered the data according to the group of participants (intervention and control), age ranges, and gender. The statistical program SPSS^®^ (version 25.0) was used. Shapiro–Wilk normality tests and equality of variances were applied using Levene’s test (selection of corresponding statistical tests, sample size of fewer than 50 participants; check if a continuous variable follows a normal distribution). Additionally, the Student’s t-test for independent samples (normal distribution; parametric test; comparative analysis; 2 different groups; mean; differences between sex and group) and the Mann–Whitney U-test (non- normal distribution; non-parametric test; comparative analysis; 2 different groups; median; differences between sex and group), the Wilcoxon signed-rank test (non-normal distribution; non-parametric test; medians; comparison of baseline, and final measurements in the same group), and the Student’s t-test for related samples (normal distribution; parametric test; mean; comparison of baseline and final measurements in the same group) were performed. Finally, ANOVA for repeated measures (normal distribution; parametric test; intrasubject; mean; comparison of weeks of intervention) and the Friedman test (non-normal distribution; non-parametric test; medians; intrasubject; comparison of weeks of intervention) were used. Additionally, Spearman’s rank correlation test (monotonic association between 2 variables, arsenic and fluoride) was used to search for correlations between variables. The significance level used was *p* < 0.05.

To measure the effect of the intervention, the number needed to treat (NNT) was calculated with *p* < 0.05 and IC 95%. The NNT was calculated using the following formula:NNT = 1/RAR(3)
where RAR is the Absolute Risk Reduction.

## 3. Results

### 3.1. Anthropometric Assessment

Anthropometric variables were similar at the start of the study for all participants, as observed in previous research. However, some notable changes in anthropometric indicators over time were identified. Regarding the indicator body weight (PC) and the percentage of fat mass (MG), no significant difference was observed at the beginning of the study or in subsequent weeks between the treated group (G1) and the untreated group (G2) (*p* > 0.05), according to the Mann–Whitney U test. However, the abdominal circumference (CA) showed differences at week 4 between groups (*p* < 0.05), with lower values observed in G1. On the other hand, differences were observed in the fat mass and abdominal circumference indicators during the study in each group, except for body weight, where only G1 presented changes over time (decrease 1–4 weeks). The results obtained showed a reduction in G1 of PC, MG and CA at the end of the intervention (Medians: PC = 46; MG = 17.50; CA = 70) in relation to the initial values (Median: PC = 46.50; MG =18.10; CA = 73). Thus, differences were obtained throughout the supplementation period (Weeks 1–4) according to the Friedman test (PC: X^2^ (2) = 22.48 *p* < 0.001; MG: X^2^ (2) = 26.78 *p* < 0.001, CA: X^2^ (2) = 46.64 *p* < 0.001) (Figure 2a).

In the case of G2, there were also differences over time for MG and CA. However, these changes were due to increases in the indicators MG (Median: S0 = 22.65; S4 = 22.75) and CA (Median: S0 = 77.50; S4 = 77.65); thus, G2 presented significant differences for these indicators but not for PC, according to the Friedman test (PC: X^2^ (2) = 2.03 *p* > 0.05; MG: X^2^ (2) = 10.76 *p* < 0.02; CA: X^2^ (2) = 24.02 *p* < 0.001), showing stability for the latter during the study (Figure 2b). It is important to note that a reduction in weight was observed at week 3, but this weight was regained at the end of the study in this group. On the other hand, considering the sex of the participants for each group, no differences were observed in the anthropometric indicators between boys and girls in G1 and between boys and girls in G2 at the end of the study, according to the Mann–Whitney U test (*Week 4: G1 p > 0.05; G2 p > 0.05*). The decrease or increase in value for each indicator can be seen in more detail in Figure 2.

### 3.2. Dietary Evaluation

Nutrient intake was similar at the beginning and at the end of the study for both groups (G1, G2), but there were some differences between boys and girls. The recommended daily requirements for some nutrients were satisfied in the male group, as also observed in previous research. Regarding the quantification of the nutrients of interest in this study, it was observed that the contents of protein (>34 g/day) and vitamin B12 (>1.8 mcg/day) ingested were adequate in boys and girls from both groups; however, the values for vitamins A (>600 mcg/day) and C (>45 mg/day) and for minerals such as iron (Fe) (>8 mg/day) and magnesium (Mg) (>240 mg/day) were only adequate for the male group. Regarding vitamin E (<11 mg/day), calcium (Ca) (<1300 mg/day), and phosphorus (*p*) (<1250 mg/day), the minimum requirements for boys and girls were not met according to the dietary intake references by population group. On the other hand, regarding food sources, it was observed that protein, Ca, Mg, and vitamins E and B12 were acquired by ingesting dairy products and milk. Additionally, other sources of protein and Fe were meat (mainly red meat) and eggs. The presence of *p* was mainly due to the intake of legumes, meat, and cheese (Appendix A). The intake of antioxidants from vegetables and green leafy vegetables was minimal due to their low acceptability and use in the preparation of salads and an emphasis on consumption with accessories (oils and fats) and through frying culinary methods. On the other hand, a high frequency of consumption of sweet fruits, such as oranges and bananas, was identified. Finally, a portion of energy was provided by ultra-processed foods. These products additionally increased the intake of critical nutrients, such as sodium (Na) (>2000 mg/day), sugars (>30 g/day), and saturated fats (>10% kcal; >17 g/day) (Appendix A).

### 3.3. F^−^ Concentration in Drinking Water and Urine

The concentrations of F^−^ in water from the school were >3.0 ± 1.72 mg/L. On the other hand, the level in household piped water, and filtered water was 2.8 ± 2.36 mg/L and 0.9 ± 1.25 mg/L, respectively. These values remained unchanged throughout the study. Regarding the concentration of F^−^ in urine, Table 1 and Figure 3 show the levels (median, minimum and maximum values), grouped by group and week of intervention. The results by sex can be seen in the supplementary materials (Appendix A). The values analyzed by sex (Table 1) (Appendix A) and by study group (Figure 3) showed significant differences from week 2 in the treated group (*p* < 0.05; *p* < 0.001). The Wilcoxon signed-rank test identified significant differences between weeks of supplementation (*S0* vs. *S1: p < 0.05; S1* vs. *S2: p < 0.001; S3* vs. *S4: p < 0.05; S0* vs. *S4: p < 0.001*) except for S2 vs. S3 (NS) (Figure 3).

Finally, the Friedman test confirmed that F^−^ levels increased during the recovery process after treatment (Median = 0.52 mg/g creatinine) in relation to the measurements made before starting supplementation (Median = 9.23 mg/g creatinine). An improvement was observed from week 1 (Median = 8.39 mg/g creatinine), but a significant difference occurred in week 2 (Median = 0.73 mg/g creatinine) and continued until the end of the treatment period for the participants who took the supplement (*X^2^* (2) = 50.98, *p* < 0.001). The untreated group did not present differences in values between weeks 0, 1, 2, 3, and 4 (NS), but a slight change was observed when comparing weeks 0 and 4 (*p* < 0.05) (Figure 3).

The Spearman test was performed for weeks 2, 3, and 4 in the group of treated participants due to the fact that statistically significant differences were observed between study groups at those times, in addition to the fact that from week 2, a notable change was observed in the two variables in the group under supplementation. However, when applying the statistical method, it was found that there was no correlation between the levels of F^−^ and As in the urine when analyzed by week of supplementation, as no statistical significance was found (*p* > 0.05). Additionally, a dispersion graph was constructed to visualize the behavior of variables F^−^ and As for week 2 (Appendix A), confirming the absence of a correlation between the variables.

### 3.4. Number Needed to Treat (NNT)

To measure the effect of supplementation, the Number Needed to Treat (NNT) was calculated, resulting in NNT= 2 (CI 95% = 0 to 2). The model was adapted to show those exposed to supplementation and those not exposed to it. The gradual and constant reduction of urinary fluoride levels at four weeks through supplementation was considered a reduction in risk, and no reduction or even an increase was considered an increase in risk. This indicated a 92% risk reduction in supplemented participants.

## 4. Discussion

### 4.1. Drinking Water (F^−^)

It is important to contextualize the study by Monroy-Torres et al. [10], where water with high levels of, firstly, arsenic (As) and then fluoride (F^−^) was identified in this community [5,10]. Currently, the values reported in this study show fluoride (F^−^) concentrations in drinking water of >3.0 ± 1.72 mg/L, a value that is in accordance with the results of previous investigations [10]. The concentrations of F^−^ as well as those of As [5], continue to exceed the established limits [32]. This situation is alarming since these elements have been recognized not only for their toxicity but also for presenting a strong positive correlation [37,38]; that is, participants exposed to high concentrations of As will also be exposed to high levels of F^−^, increasing the risk of presenting fluorosis and arsenicosis at the same time, a phenomenon that was observed with the exposure biomarkers used for As [5,10] and F^−^ (Table 1) and confirmed by dental clinical examination in the groups included in this study (Figure 4). A qualitative survey identified the types of water used for drinking and food preparation. The participants who responded affirmatively to using tap water for drinking (including tap water from the school) and food preparation presented severe cases of fluorosis (Figure 4).

The community is aware of the concentrations reported in this investigation, as are the health authorities, especially the Jurisdiction of Irapuato. In fact, a water treatment plant was in place prior to this, and other studies were conducted, but to date, there is still no ability to exercise the right to water in the State of Guanajuato or in many parts of Mexico and the Latin American region. Therefore, although there is a treatment plant, it is easier for the population to consume piped water. This is why, in addition to informing the authorities and promoting communication activities in the population, this research aimed to contribute to the promotion of adequate nutrition, promote alternatives for access to nutrients and an adequate state of health, and reduce negative health effects by influencing the toxicokinetics of this type of contaminant.

### 4.2. Fluoride in Urine (F^−^)

In this investigation, it was observed that the level of urinary F^−^ in the supplemented group was greater than that in the control group. In the second week, in the supplemented group, the urinary F^−^ levels presented a difference, showing a possible recovery process (Baseline: 9.23 mg/g creatinine; Week 2: 0.73 mg/g/creatinine; Week 4: 0.52 mg/g creatinine), but the changes in the control group were not significant (Baseline: 8.17 mg/g creatinine; Week 2: 7.86 mg/g/creatinine; Week 4: 7.89 mg/g creatinine) throughout the study (Figure 3) (Table 1) (*S0* vs. *S1:* p *< 0.05; S1* vs. *S2: p < 0.001; S3* vs. *S4: p < 0.05; S0* vs. *S4: p < 0.001*). Some authors have reported similar findings after applying corrective measures based on a diet rich in Ca, Fe, vitamin C, vitamin B12, antioxidants, and micronutrients (4–6 weeks) for the mitigation of fluorosis [25]. The authors observed a reduction in the concentration of urinary F^−^ (2.65 mg/L to 0.835 mg/L) after the intervention practices. Other studies have reported that, after supplementation with Ca, vitamin D, and vitamin C, a significant reduction in serum fluoride was observed, and F^−^ in urine increased with pre-supplementation values of 9.95 ppm, 13.3 ppm, and 15.0 ppm, and post-supplementation values of 10.9 ppm, 14.64 ppm, and 17.06 ppm, respectively [39]. The significant changes observed in the treated group after supplementation with these micronutrients may have been due to the fact that a diet high in calcium can contribute to the net secretion of fluoride in the gastrointestinal tract [39,40] and can reduce the degree of F^−^ absorption [41]. In fact, the presence of calcium in the intestine directly affects the absorption of fluoride ions and also improves the serum calcium level [22], as well as increasing renal elimination [39]. Additionally, calcium, vitamin C, and vitamin D supplementation have definite roles in improving renal clearance [39]. Vitamin D, in low doses, enhances calcium absorption and retention without causing hypercalcemia and thus directly affects fluoride ion absorption [22]. Calcium and vitamin D reduce compromised renal perfusion by correcting secondary hyperparathyroidism induced by a high fluoride intake [39]. On the other hand, it has been reported that reactive oxygen species play an important role in the development of endothelial dysfunction of the renal vasculature and that vitamin C, being an antioxidant, improves renal clearance under conditions of oxidative stress, a situation involved in subjects exposed to F^−^ [39].

On the other hand, it has been reported that diets rich in antioxidants and minerals (Ca, Mg, K, and Al) from vegetables, fruits, and dairy products have the ability to improve F^−^ excretion and promote recovery in participants exposed to the toxin [24,25,41]. It is well known that approximately 80–90% of orally ingested F^−^ is absorbed in the gastrointestinal tract by passive diffusion and that up to 40% of ingested fluoride can be absorbed in the stomach, where the extent of absorption is inversely related to the pH of the stomach contents [24]. Due to pH dependence, gastric absorption, distribution, and renal excretion of F^−^ could be affected by changes in the body’s acid–base balance. It is for this reason that the diet influences the acid–base balance through the supply of acid and alkaline precursors of food, which alkalinize or acidify the urine [24,41,42]. Protein-containing foods such as meat, grains, and dairy products [24,41] generate H+ ions through hepatic oxidation of the Sulfur-containing amino acids cysteine and methionine. However, most fruits and vegetables (vegetarian diets) act as basic precursors, as well as providing large amounts of Mg and K in the diet [24,41]. Therefore, the consumption of fruits and vegetables could lead to an increase in urinary pH, increasing the renal excretion of F^−^ and, consequently, decreasing the concentration of fluoride in the body [24]. On the other hand, although fluoride permeation across the gastric mucosa is dependent on the acidity of the stomach contents, fluoride absorption in the small intestine is independent of pH and occurs largely as fluoride ions cross the permeable epithelium through the narrow paracellular junction channels between epithelial cells. Therefore, ions such as calcium, magnesium, and aluminum form insoluble complexes between the fluoride anion and multivalent cations in the alkaline small intestine, significantly reducing fluoride uptake by the bones and teeth [24,43]. Additionally, according to Zheng et al. [44], Ca and P interfere with the gastrointestinal absorption of F^−^. Additionally, another nutrient, boron, improves the elimination process [45]. Other studies have observed that supplementation with adequate dietary factors (consumption of green leaves) such as animal and vegetable proteins [40], calcium, magnesium, melatonin, and selenium can significantly reduce the toxic effects of F^−^, decreasing its absorption and increasing the removal process from the body [45].

In the present study, differences in urinary F^−^ values were observed (*S0* vs. *S1: p < 0.05; S1* vs. *S2: p < 0.001; S3* vs. *S4: p < 0.05; S0* vs. *S4: p < 0.001*) (*X^2^* (2) = 50.98, *p* < 0.001), as were differences in the severity of dental fluorosis between girls and boys in the supplemented group, where girls presented lower levels of F^−^ (Table 1) and a lower degree of dental fluorosis (Figure 4). According to Mann et al. [46], boys are more susceptible to F^−^ retention, and the severity of fluorosis tends to be greater. However, another study with 479 participants from towns in northwestern Mako found no significant differences between boys and girls [47], showing that the results do not depend on sex but on nutritional habits and dietary patterns in addition to the excessive concentration of F^−^ in piped water [40,47]. On the other hand, the fluoride content of the body is not under physiological control, and urinary excretion will depend on various factors [48]. In addition to diet, there are different variables that can influence fluoride metabolism and, therefore, the variation in F^−^ levels in fluids. The most important are acid–base disorders, hematocrit, altitude, physical activity, the circadian rhythm, hormones, body physiology, renal function, genetic predisposition [25,49], age, amount, duration of exposure [48], nutritional status (malnutrition, obesity), and systemic diseases that affect the physiological status of an individual [25,50,51,52]. For this reason, there will probably always be variation between different age groups, individuals, and populations.

On the other hand, regarding the relationship between intelligence quotient (IQ) and the presence of fluorosis and arsenicosis in boys and girls, the evidence suggests that fluoride (F^−^) and arsenic (As) may adversely affect IQ. According to Rocha-Amador et al. [9], in a study sample of 132 children aged from 6 to 10 years old and living in rural communities in San Luis Potosi, Mexico, where there are high levels of F and As in the drinking water, an inverse association was observed between F in the urine and Performance, Verbal, and Full IQ scores (β values = −13, −15.6, −16.9, respectively), and a significant association was observed between As in the urine and Full IQ scores (β = −5.72, *p* = 0.003), suggesting that children exposed to either F or As have an increased risk of having a reduced IQ score. Although our objective was not to measure IQ, the informal interviews with high school teachers showed that young people had learning problems; hence, we recommend that these findings should be evaluated in detail in future studies to expand our results on this problem.

The connection of fluoride with iron deficiency anemia is also important. The results of the previous study with the same group of participants exposed to As [5], which were confirmed in this study, for F^−^ (>3 mg/L) show that some participants had concentrations lower than 12 g/dL of Hb at the beginning of the investigation. Other investigations reported low Hb values at the beginning of the study in subjects exposed to F^−^ (13 g/dL) [25]. In addition, there were cases of anemia in the participants (<12 g/dL) [9]. Some investigations have mentioned that the main cause of anemia is due to contamination by F^−^ in drinking water. Excessive intake of F^−^ leads to anemia due to poor absorption of nutrients, including iron, demonstrating that the presence of anemia is the result of F^−^ toxicity [53]. Fluoride has been shown to cause temporary destruction of the mucous membrane and induce loss of microvilli in the intestinal mucosa, leading to a decreased absorption of micronutrients. This directly results in poor absorption of iron [53,54]. On the other hand, it has been reported that toxic chemicals (F^−^) destroy intestinal microorganisms capable of producing vitamin B12, resulting in the deficiency of this vitamin and folic acid, essential for hemoglobin biosynthesis, and consequently inducing the appearance of anemia [53].

However, despite the role of this environmental contaminant in the non-absorption of nutrients, research by Santiago-Saenz et al. [5] reinforced the findings of Monroy-Torres et al. [10] that after an intervention with antioxidants (phenolic acids and flavonoids, vitamin C), protein, and micronutrients, similar to the use of a formulation based on quelites [29], the Hb concentration increased above 1.65 g/dL (Supplemented: Week 0, 13.47 g/dL; Week 2, 15.12 g/dL). Some studies have similarly observed increases in the Hb concentration (13.7 to 15.4 g/dL) and a decrease in the number of cases of participants with anemia (>12 g/dL) after nutritional intervention practices with foods rich in antioxidants (polyphenols), proteins, carbohydrates, and vitamins (C, B12) through dairy products, cereals, vegetables, and fruits [25,53].

On the other hand, Santiago-Saenz et al. [5] reported elevated values of MDA (malondialdehyde) in the groups of participants exposed to As (baseline measurements). The MDA results observed in both study groups (Control: 2.88; Experimental: 3.01 µM/g creatinine) (pre-supplementation) confirm the ability of these elements to induce significant increases in MDA, indicating that our study participants (same participants exposed to As) had high levels of oxidative stress due to prolonged exposure to both pollutants. Several studies have identified oxidative stress due to the presence of F^−^ in animals and humans [39]. Ailani et al. [39] mentioned that oxidative stress increases malondialdehyde levels in children with endemic skeletal fluorosis. On the other hand, other investigations have shown that the combined exposure to arsenic and fluoride causes oxidative damage in the rat brain and also decreases the activity of antioxidant enzymes and increases lipid peroxidation [45]. Arsenic and fluoride are known to increase lipid peroxidation and inhibit antioxidant enzymes in the liver and kidneys of fluorinated and arsenified animals. It is often proposed that the enhanced production of active oxygen is a manifestation of membrane disruption caused by bilayer oxidation of polyunsaturated fatty acids, a process known as lipid peroxidation [55].

However, the literature mentions the importance of the use of antioxidants, vitamins, and essential elements as the treatment for chronic fluoride toxicity. It was reported by Santiago-Saenz et al. [5] that MDA levels can be reduced (>50%) after supplementation with quelites (Supplemented: Week 0, 3.01 µM/g creatinine; Week 4, 0.98 µM/g creatinine). Other studies observed that, after a 6-month supplementation period with calcium, vitamin D, and vitamin C, the value of superoxide dismutase (SOD) decreased [39]. On the other hand, after vitamin E supplementation in mice co-exposed to arsenic and fluoride, the recovery of altered antioxidant enzymes and depletion of reactive oxygen species (ROS) was observed [56].

Based on these results, vitamins C and E reverse the toxic effects induced by fluoride exposure. The mechanism of action of vitamin C could be its powerful reducing action. Under conditions of fluoride toxicity, the generated free radicals attack the double bonds of polyunsaturated fatty acids, starting a chain reaction and affecting membrane integrity and cell function. This chain reaction is inhibited by vitamin E (α-tocopherol) by reacting with free radicals and being converted to a harmless α-tocopheroxyl radical. This α-tocopheroxyl radical, thus formed, is converted back to α-tocopherol by cytosolic vitamin C. Therefore, vitamins C and E show a synergistic action in the recovery variables altered by oxidative stress and organ damage due to fluoride exposure [45]. Additionally, regarding vitamin E, the protective effect against toxicity induced by arsenic and fluoride could be attributed mainly to its antioxidant properties or its location in the cell membrane and its ability to stabilize the membrane by interacting with the fatty acid chain.

Vitamin E can be useful for restoring altered biochemical variables, showing its use as a potential free radical scavenger. These properties can allow the preservation of cell membrane functions, including ion transport and membrane fluidity. It can also prevent the release of Fe^2+^ and Mg^2+^ from their binding proteins, which could decrease the rate of lipid peroxidation [56]. On the other hand, flavonoids such as quercetin have also been studied to explore their effects against fluoride intoxication and have been shown to have a beneficial role in reducing lipid peroxidation following intoxication by this pollutant. The antioxidant action of quercetin is due to the presence of hydroxyl groups in ring B of the molecule, which contain two catechol or three pyrogallol hydroxyls [45].

The number needed to treat (NNT) was 2 with an IC 95% from 0 to 2; that is, by treating two people exposed to fluoride in the water with this supplementation, a reduction in risk would be observed. However, this should be studied further since the interval was not reliable due to the sample size and the variability in the results. Thus, the efficacy of supplementation should continue to be evaluated in future study designs.

Finally, these findings demonstrate the actions of the essential nutrients that the supplement provided to the treated group [5]. The supplement was rich in various mineral compounds (Ca, Mg, B, and P) and antioxidants (phenols and flavonoids, including quercetin and rutin), vitamin C, protein, and amino acids. Together with the consumption of fruits, vegetables, cereals, beans, and foods of animal origin (Appendix A), it was able to promote recovery from exposure to F^−^ and increase the level of F^−^ in the urine, allowing subsequent recovery. The authors explained that urine F^−^ levels need to be reduced for recovery [25]. However, despite the fact that this study presented information related to the excretion of F^−^ in the urine, the sample size and the conformation of the control group were limitations. However, given the observed effects, the study has strength; therefore, similar results should be observed in other populations when replicating this study. On the other hand, it is recommended that future investigations monitor the level of physical activity and identify the temporality and level of education of the parents and family income. With regard to adherence, it is recommended that information and family and community support are integrated so that there is clarity about the use, storage, benefits, and interactions of supplements. Additionally, we must consider the role of the microbiome as the first metal detoxification mechanism, as it has been reviewed by some authors [57]. In addition, future investigations should monitor fluoride values in drinking water and food intake in participants, as well as measure elimination in feces in addition to urine, although, as mentioned, urine is the best indicator of the level of excretion.

## 5. Conclusions

Supplementation (4 g) of quelites (*Portulaca oleracea* and *Chenopodium berlandieri*) for 4 weeks was associated with a gradual and constant (weekly) reduction in urinary fluoride levels, which was observed from the second week onwards. Although the NNT was 2, the confidence interval of 0–2 shows that the efficacy of supplementation should be measured by increasing the sample size. With the findings found regarding the reduction of risk between cases and controls, we observed a 92% reduction from baseline to final values in the intervention group (cases).

Therefore, this study provides information on treatment options and alternatives against F^−^ based on local foods (quelites) for the Mexican population. With these findings and previous experiences, it is recommended, in addition to the demand for the right to water, that this intervention is integrated into the rights to food and health.

## Figures and Tables

**Figure 1 foods-11-03071-f001:**
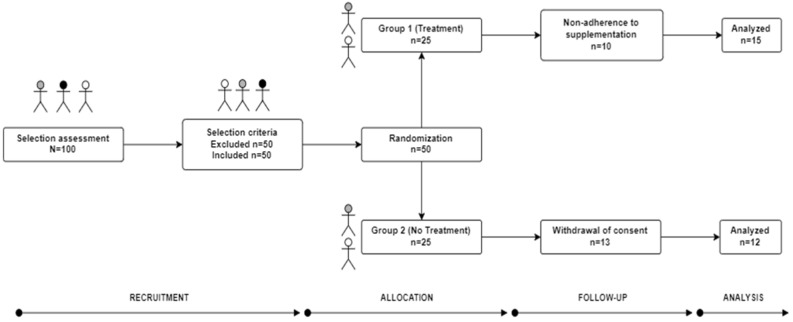
Diagram of the clinical study design (COSORT).

**Figure 2 foods-11-03071-f002:**
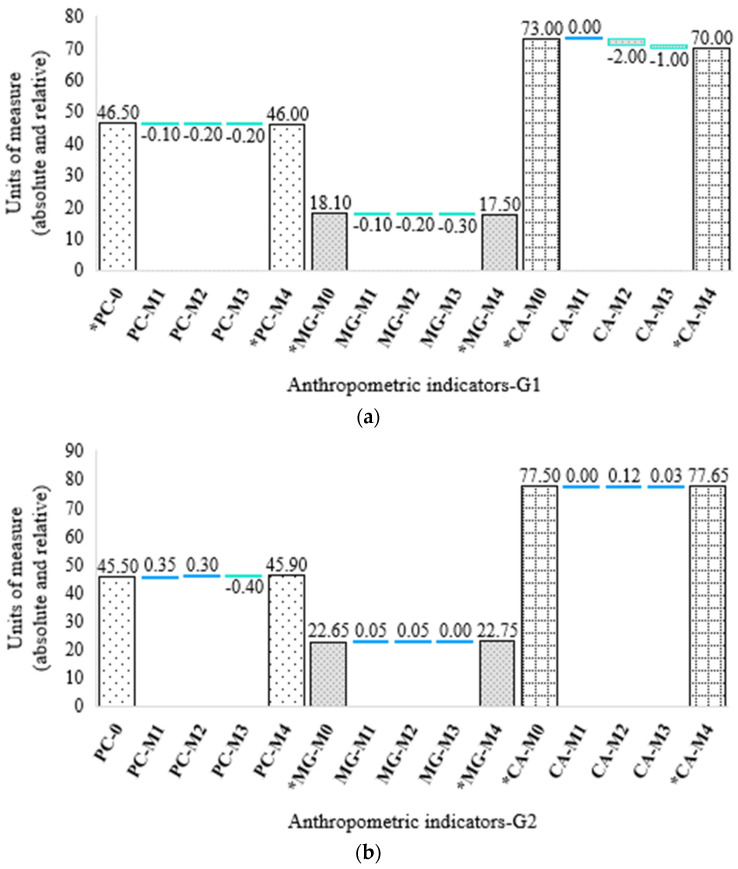
Increases and decreases in anthropometric indicators. (**a**) Treated group (G1); (**b**) Untreated group (G2). The bars with the same texture represent the baseline and final measurement of an evaluated indicator; the blue and green lines show the increase or decrease in each indicator throughout the 4−week period. Body weight (PC); Fat mass (MG); Abdominal circumference (CA); Measurements 0−4 (M0−M4). Absolute measurement units (kilograms and grams for PC and centimeters for CA) and relative values (percentage for MG) are given. * Statistically significant difference according to the Friedman test (*p* < 0.001).

**Figure 3 foods-11-03071-f003:**
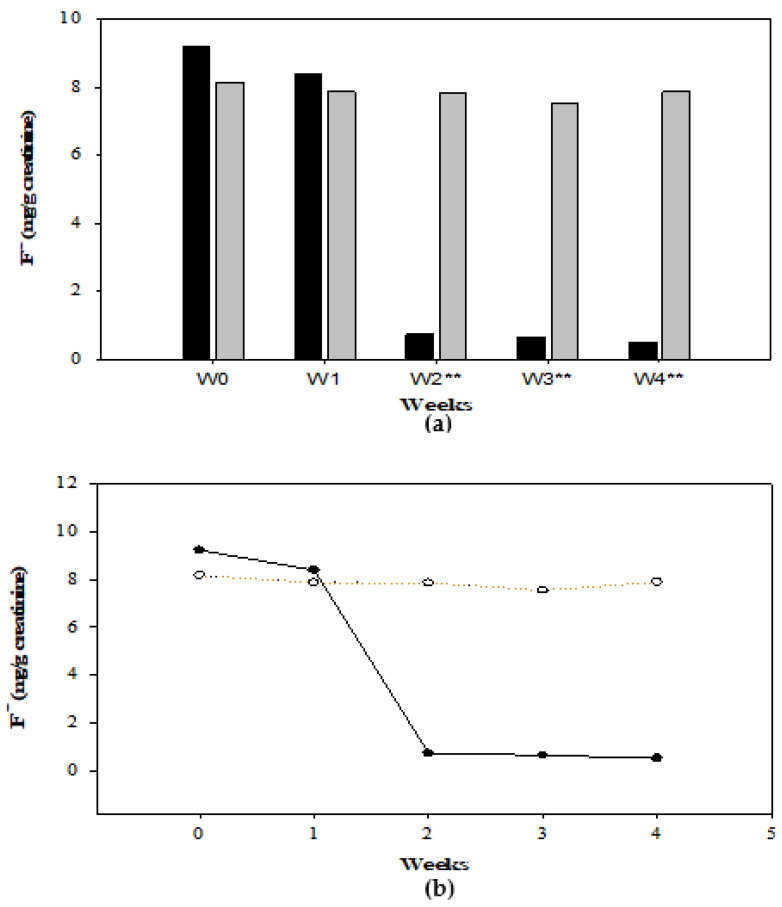
(**a**) Comparison of the F^−^ concentration (median) before, during, and after 4 weeks of supplementation between the treated group (G1−Black bars) and the untreated group (G2−Gray Bars) (W0: baseline; W1: week 1; W2: week 2; W3: week 3; W4: week 4). ** Statistically significant difference between groups according to the Mann–Whitney U test with *p* < 0.001. (**b**) Comparison of the concentrations of F^−^ (median) in the treated (Black line) and untreated groups (Dotted line) by week of treatment. Friedman and Wilcoxon tests were performed to determine differences between weeks for G1 and G2. The interquartile range (IQR) was included for W0 (G1: 2.43; G2: 2.93), W1 (G1: 3.52; G2: 2.50), W2 (G1: 0.44; G2: 2.40), W3 (G1: 0.70; G2: 1.60), and W4 (G1: 0.34; G2: 2.36).

**Figure 4 foods-11-03071-f004:**
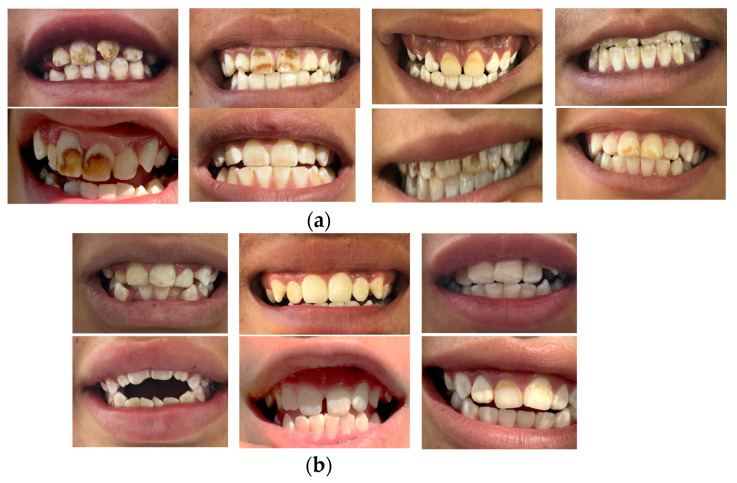
Front teeth of participants who presented high concentrations of F^−^ and urinary As at baseline (week 0) of the investigation. (**a**) Boys; (**b**) girls. The boys and girls who presented high contents of the biomarkers frequently consumed tap water for drinking and food preparation.

**Table 1 foods-11-03071-t001:** Concentrations of urinary F^−^ in the untreated group and in the group treated with the supplement for 4 weeks.

	G1 Total Sample *n* = 15	Boys *n* = 9	Girls *n* = 6		G2 Total Sample *n* = 12	Boys *n* = 4	Girls *n* = 8	
Weeks	F^−^ (mg/g Creatinine)							
				*p*				*p*
0	9.23	9.42	9.01	NS	8.17	8.61	7.55	NS
1	8.39	8.21	8.48	NS	7.87	8.29	6.64	NS
2	0.73	0.79	0.53	<0.05	7.86	8.08	7.16	NS
3	0.64	0.69	0.46	<0.05	7.55	7.73	7.46	NS
4	0.52	0.68	0.43	<0.05	7.89	8.16	7.09	NS

The values are the median concentrations of F^−^ (mg/g creatinine) in the treated group (G1) and untreated group (G2). Week zero represents the baseline measurement of the corresponding study group (NS: no significant difference between males and females in the same group per week according to the Mann–Whitney U test).

## Data Availability

Not applicable.

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
