# Peer review of "Fluoride Concentration in Urine after Supplementation with Quelites in a Population of Adolescents"

_foods, 2022, doi:10.3390/foods11193071_

Round 1

Reviewer 1 Report

Your paper is very important, but is missing several key points.  Where is the data on total fluoride intake from water and the foods? What is the distribution of fluoride levels, with and without the quelites, in both urine and feces? 

Reviewer 2 Report

In the present manuscript, the authors presented effect of supplementation with quelites on fluoride concentration in urine in a population of adolescents. The topic is interesting and and the results are supported by presented data.

In my opinion, the work is extremely similar to the previous one in which the study with arsenic was performed on the same population. In that sense, there are not too many novelties. On the other hand, the methodology and results are clearly presented.

I suggest authors to correct part in conclution: "the supplement promoted the increase in hemoglobin levels and the reduction og malondaldehyde (MDA)". I suggested authors to exclude this sentance from the conclusion because it was not part of work described in this Manuscript.

I recommend this manuscript for publication after minor revision.

Round 2

Reviewer 1 Report

This is a very important paper and topic to study, but it was not well presented.  It appears that ref 5 was originally published and that you are including MDA and Hb data from that souce, which is fine, but the two articles should show the data.  In the conclusions, it is stated that quelites increase hemoglobin levels and reduce MDA. Where is this data presented?

Do you conclude that quelites promote lower urinary fluoride by enhancing fluoride elimination in feces (ie less uptake) or where is the fluoride going?  Fluoride intake, elimination has not been balanced...

You must be consistent in your labels for Figure 2.. Is AC the same as CA? Please use the same lettering order.

I think Figure 3 is not necessary and should be eliminated, or put in as a supplemental figure.

Explain what is meant by "purified water," How was the fluoride reduced to 0.9 mg/L?

Figure 4 A and B requires error bars.

Section 4.2 the authors state "urinary F- in the supplemented group INCRESED",

Figure 5 needs to include the details concerning the actual Fluoride values in drinking and food preparation...

but in the next sentence the authors state, "urinary F- levels were significantly reduced"  Fix this discrepancy!

Again, the IQ effects for children with that level of dental fluorosis should be emphasized.
